# Lignin-Based Nanocarrier for Simultaneous Delivery of ^131^I and SN-38 in the Combined Treatment of Solid Tumors by a Nanobrachytherapy Approach

**DOI:** 10.3390/ph18020177

**Published:** 2025-01-27

**Authors:** Aleksandar Vukadinović, Miloš Ognjanović, Milica Mijović, Bryce Warren, Slavica Erić, Željko Prijović

**Affiliations:** 1“Vinča” Institute of Nuclear Sciences-National Institute of the Republic of Serbia, University of Belgrade, 11351 Belgrade, Serbia; 2Institute of Pathology, Faculty of Medicine, University in Priština-Kosovska Mitrovica, 38220 Kosovska Mitrovica, Serbia; milica.mijovic@med.pr.ac.rs; 3Natural State Science LLC, 415 N. McKinley Street, Little Rock, AR 72205, USA; 4Faculty of Pharmacy, University of Belgrade, 11221 Belgrade, Serbia; slavica.eric@pharmacy.bg.ac.rs

**Keywords:** cancer, ^131^I, radioisotope, SN-38, nanobrachytherapy

## Abstract

**Background:** The rapid rise in cancer incidence significantly augments efforts to improve cancer treatments. A multimodal approach in the nanobrachytherapy of solid tumors is one of the promising methods under investigation. This study presents a novel biocompatible lignin-based nanomaterial, loaded with cytostatic agent SN-38 and radionuclide ^131^I, for simultaneous radiation and chemotherapy of solid tumors by a nanobrachytherapy approach. **Method:** Nanoparticles of ~100 nm in size, composed of lignin alone or loaded with 10% (m/m) of SN-38 (SN-38@lignin), were synthesized using a bottom-up approach and characterized. Subsequent radiolabeling of the nanoparticles by ^131^I produced ^131^I-lignin and ^131^I-SN-38@lignin. Their antitumor efficiency was tested against luciferase-expressing 4T1 mouse breast cancer xenografts of ~100 mm^3^ size on Balb/c mice. **Results:** An intratumoral injection of 1.85 MBq of ^131^I-lignin was retained within the tumor and achieved a moderate twofold decrease in tumor size compared to the control group. Injecting SN-38@lignin containing 25 µg of SN-38 decreased tumor size 3.5-fold. The therapy using the same doses of ^131^I-SN-38@lignin produced the most potent antitumor effect, with tumors being 6-fold smaller and having extensive intratumoral necrosis, all of it without signs of systemic toxicity. **Conclusions:** These results support the intratumoral delivery of lignin-based nanomaterial carrying radioisotopes and camptothecins for effective multimodal anticancer therapy.

## 1. Introduction

Despite enormous century-lasting efforts, cancer incidence is steadily growing, while cancer is still one of the most frequent causes of death worldwide [1]. The three traditional pillars of anticancer therapies (radiotherapy, chemotherapy, and surgery—alone or combined) are not enough to cope with the surge, let alone to conquer the disease. A significant number of more advanced therapies based on biological approaches (cell-based, immune system-based, or gene therapies, for example) or advanced technology have been under investigation [2,3,4,5,6,7,8,9,10]. Even being effective in special cases, the cost and complexity of the approaches make them of limited use for broader communities worldwide. Consequently, the traditional techniques are still the main saviors of life when cancer therapy is in consideration. Improvement of the classic therapies in order to preserve their positive sides but avoid the side effects and pitfalls seems a reasonable route for the successful development of an effective yet affordable cancer therapy. Several traditional methods but with modern modifications and improvements are under investigation, while some of them are already in clinical use. For example, a significant advance in chemotherapy is achieved by developing targeted chemotherapy and nano-therapies [11,12,13]. Similarly, recently, there have been attempts to modernize one of the oldest approaches of radiotherapy, brachytherapy.

Classic brachytherapy is a form of radiotherapy where a radioactive source in the form of a solid material termed “seed” is surgically implanted into a tumor, effectively irradiating it with very limited damage to the surrounding tissue [14,15]. Its relatively low cost, high effectiveness, and relatively low invasiveness make it the therapy of choice for several cancer types (head and neck, breast, cervix, and eye), but most traditionally for prostate cancer. However, a few drawbacks caused the decrease in its application. First, the production of the radioactive solid “seeds” is costly, the choice of radioisotopes is limited, and the surgical implantation is technically demanding and requires a second surgery to remove the seeds once the therapy ends. To keep the benefits but avoid the drawbacks of brachytherapy, a new approach named nanobrachytherapy (NBT) is in development.

Nanobrachytherapy is based on injecting isotope-laden radioactive nanomaterial instead of solid “seeds”, aiming to be a more effective and simpler therapy approach than brachytherapy itself [16,17,18,19,20,21]. The expected benefits compared to traditional brachytherapy are the broader choice of radioisotopes and doses, ease of application, lack of need for removal from the tumor after the treatment, and the possibility to combine it with other anticancer agents. All of these factors make NBT a flexible multimodal therapy platform that can be easily adapted to individual tumors. Moreover, intratumoral injection has steadily gained popularity in the last decade, with a plethora of materials, from small drugs to biopolymers to nanomaterials, being delivered this route.

Many different nanomaterials for cancer therapy have been generated up to date, while several have been in human use already [22,23,24,25,26]. Among nanomaterials for nanobrachytherapy of solid tumors reported to date are several we produced that were composed of superparamagnetic iron oxide-based nanoparticles (SPIONs) [17,18,21], combined with proven radionuclides used in human diagnostics and therapy, like ^131^I, ^90^Y, and ^177^Lu. When bound to iron oxide-based SPIONs, which can generate heat when in an oscillating magnetic field [27,28,29,30,31,32,33,34,35], they may combine radiotherapy with hyperthermia to better the antitumor effect. Further, we and others included antibodies against tumor-specific antigens in the radioactive nanomaterial to ensure its long-lasting retention in the place of injection and minimize leakage [21]. All the experiments showed excellent antitumor effects, firmly supporting the nanobrachytherapy approach as a promising route of investigation for an effective yet simple and affordable solid cancer therapy approach.

The use of an iron oxide-based matrix as a carrier for combined radiotherapy and hyperthermia, even though it has shown good effects, may not be an optimal choice for several reasons. First, although it is excellent in many instances, iron-based material is neither exactly biocompatible nor easily degradable. It raises the question of eventual toxicity and immune reactions, as described in some cases [36,37]. Second, hyperthermia, even proven to contribute to the anticancer effect, did not complement radiation to the degree we expected. Furthermore, no instrumentation is readily available on the market for generating magnetic hyperthermia for human therapy. All of this urged us to search for a more suitable material as a carrier for NBT.

Aiming for an affordable yet biocompatible material capable of being chemically modified to carry on other materials and to be radiolabeled as a base for the development of our new NBT therapeutics, we opt for biopolymers. Biopolymers are a broad group of materials that can be used for nanomedicine, starch, cellulose, pectin, chitin, and lignin being just a few of them [38]. All of them are abundant, readily available, and economically affordable, yet most of them are biocompatible, biodegradable, and ecologically friendly [38,39,40,41,42,43,44,45,46,47,48,49,50,51,52,53,54,55]. Additionally, they possess suitable groups to be radiolabeled. Considering all of this, we chose lignin as a material for a new series of nanomaterials for nanobrachytherapy.

Lignin is a biopolymer ubiquitously present in plant cells, making it one of the most abundant, economical, and affordable biomaterials [56]. Its chemical structure is very complicated and varies depending on the source and route of preparation [57]. Generally, it is accepted that lignin is built from three “lignols”, the monomers of aromatic alcohols named syringyl-(S), guaiacyl-(G), and p-hydroxyphenyl-(H), arranged into a tri-dimensional scaffold with methoxy-, carbonyl-, carboxyl-, and hydroxyl-groups (Figure 1A). This makes lignin structurally stable, biocompatible, and well suitable for chemical modifications, including radiolabeling [40]. Additionally, it possesses intrinsic antioxidant and anticancer activities [39]. Among several interesting properties, its natural affinity to iodine makes it a promising candidate for removing radioactive iodine from contaminated soil [41,46,58]. This ability and our previous experiments that show the abilities of the ^131^I against tumors [21] lead us to propose the use of lignin as a carrier for radioactive iodine as a therapeutic. Additionally, lignin’s chemical structure is very suitable for interacting with many chemical entities, allowing an introduction of a chemotherapeutic to augment/synergize the action of the chosen radioisotope. Knowing that the action of radiation causes DNA damage, the use of chemotherapeutic drugs acting on DNA repair looks like a reasonable option. Another requirement of the potential drug to be combined is to be chemically compatible to form adducts with lignin. Considering all of this, we opt for camptothecins as promising, highly potent drugs acting on the DNA level.

Camptothecins are semi-synthetic or synthetic derivatives of natural alkaloid camptothecin, isolated from the bark of the south-Asian tree *Camptotheca acuminata*. It is a semi-planar, aromatic, five-ring molecule with a pH-dependent lactone ring required for cytostatic activity (Figure 1B). The activity of camptothecin is based on the interaction of its lactone ring with the replication fork of cells in the S-phase, forming a stable covalent bond with a DNA strand, effectively stopping the division and ultimately leading to the cell’s death. Its high cytotoxicity makes it an excellent choice for development of anticancer chemotherapeutics. Indeed, two derivatives, Irinotecan (CPT-11) and Topotecan, are already in clinical use, primarily as first-line treatments for colon cancer, and are also employed in the treatment of various other cancer types [59,60,61,62,63,64,65,66,67]. The aromatic and semi-planar structure of camptothecins, possessing similar functional groups (hydroxy and carboxy) as lignin, indicates they may efficiently interact with lignin by non-covalent interactions, so it is reasonable to try to develop lignin nanoparticles loaded with camptothecins to be co-delivered to tumors.

CPT-11 is an inactive camptothecine prodrug, which requires a complex enzymatic conversion to generate its active form, SN-38 [68]. The released drug is 100–1000-fold more potent than CPT-11 and, being hydrophobic, enters the cells very fast (~30 s). Moreover, if it leaks into circulation, it undergoes an efficient inactivation by glucuronidation and excretion by urine and bile, thus avoiding systemic toxicity. Additionally, being a major metabolite of an approved drug, its use in humans does not require separate FDA approval. All of this suggests that SN-38 is a good candidate for intratumoral therapy to complement radiation-induced DNA damage.

Here, we describe the development and characterization of a new biocompatible lignin-based nanomaterial carrying a potent camptothecin-class cytostatic SN-38 and a proven radiotherapeutic ^131^I to simultaneously deliver radionuclide and chemotherapy to tumors by a nanobrachytherapy approach.

## 2. Results and Discussion

### 2.1. The Nanoparticle Preparation and SN-38 Loading

The lignin and mixed SN-38-lignin nanoparticles were generated by a bottom-up approach, employing dialysis to gradually exchange the suitable solvent (DMSO) with the anti-solvent (water). The procedure was developed and investigated in detail by Lubna Siddiqui et al. [40]. It was shown that pH control during dialysis is very important, and the process proceeds in two phases. First, dialysis in acidified water pH 5.2 generates the nucleation centers of lignin. Second, pure water dialysis allows the growth of the particles. Consequently, varying the dialysis times in acidic and neutral pH, as well as the starting concentrations, may influence the product’s size, shape, and polydispersity index.

To closer examine the method, we varied the parameters of pH from 4 to 6.5, the dialysis time in acid from 2 h to 6 h, and the lactone SN-38 to lignin ratio from 5% to 40% (m/m). The method indeed reliably generated lignin and lignin-SN-38 nanoparticles of size commonly ranging between 100 and 200 nm, with fairly narrow size distribution and PDI values of about 0.2–0.3 (Appendix A). To produce the material for the in vivo application, we repeated the synthesis in triplicates, with starting concentrations of SN-38 at 1 mg/mL and lignin at 5 mg/mL (20% ratio), using 4 h dialysis in pH = 5.2–5.3 followed by 20 h dialysis in water. The average size of the produced nanomaterial was 121.08 ± 54.38 and polydispersity index 0.20 for lignin alone, and 143.17 ± 66.69 and PDI = 0.21 for lignin-SN-38 (Appendix A). Even though the nanoparticle size distribution we obtained is fairly narrow within the batch and well below 250 nm, which is considered adequate for biological applications, there was a significant variation in the average size (but not in the PDI index) among batches. We attribute it most probably to the difficulties in controlling the pH of water during dialysis, a critical step in the procedure. However, every batch had an average size below 250 nm and PDI~0.2, meaning that every batch could be used in vivo.

These values are close to the reported in the original paper, where the size of obtained nanoparticles was ~110–150 nm and the PDI index 0.26–0.31 [40]. Considering that common requirements for nanoparticles to be used in in vivo therapy are that the size is below 250 nm and the PDI value between 0.1 and 0.3, our values are well within the range. The batches with a slightly smaller size of 86.48 nm for lignin and 107.00 nm for lignin-SN-38 (Figure 2A) were chosen for the in vivo therapy, expecting better intra-tumoral penetration.

The obtained material was further characterized by HPLC to determine the content of SN-38 and FTIR spectra to prove no free SN-38 presence. The results are provided in Figure 2B–D, while more details are presented in the Appendix A. Interestingly, while the loading amount increased with the SN-38 concentration (Figure 3C), the loading efficiency, as confirmed by HPLC analysis, did not vary much regardless of the amount of SN-38 present, averaging about 55% for 5–40% of SN-38 to lignin ratio (m/m) (Figure 2D), being slightly higher (65%) for lower amounts of SN-38. We presume it was a consequence of the pH-dependent re-establishment of lactone–carboxy equilibrium of SN-38, where the carboxy form is water soluble and may be dialyzed away. For in vivo application, we opted for SN-38 content close to 10% (m/m), considering the very potent cytotoxicity of SN-38. Interestingly, almost the same loading efficiency was reported earlier with 10-hydroxy-camptothecin, resulting in ~55% efficiency and 9% (m/m) ratio [69,70], further supporting the opinion that the loading efficiency is related to the structure of camptothecins and their lactone–carboxy equilibrium.

FTIR spectra for pure SN-38 and lignin closely correspond to those published. For SN-38 alone, the most interesting structural property is its lactone ring, while the most distinguishable are C=N and N-H bonds that are absent in lignin. The FTIR spectra show a distinctive C=N line at 1660.7788 cm^−1^. The area 3200–3500 cm^−1^ can be attributed to N-H bonds present. The most characteristic signature of SN-38 is probably the line at 1733.9582 cm^−1^ representing the lactone and carboxy carbonyl groups at 1657.440 cm^−1^, while the line at 1166.1860 cm^−1^ may present the carbonyl group stretching in carboxy form. The line at 3579.8486 may present the O-H stretch in carboxylate. For lignin alone, the situation is more complex because of the structural variations due to the source material differences. The line at 1021.0886 represents the O-H in primary alcohols, abundant in lignin. The line at 1130.8779 represents the C-H stretch, while the lines at 1212.8695 cm^−1^ and 1268.385 may represent the C-O stretch. The line at 1514.4196 is characteristic of an aromatic ring. The two lines at 1658.2552 and 1740.2668 are due to conjugated and non-conjugated carbonyl groups, respectively. The lines at 2952.7758 cm^−1^ represent methyl and methylene sidechains on aromatic rings. Further, the broad line centered around 1388 cm^−1^ is characteristic of aliphatic and aromatic hydroxyl groups. The spectra of the lignin nanoparticles are characteristic, distinguishable from the starting material, and virtually identical among the lignin nanoparticles obtained with 5–40% SN-38. Interestingly, despite the content of SN-38 being confirmed and determined by HPLC to be ~10% (m/m), no spectral lines of free SN-38 were observed in the nanomaterial, even though they are present when lignin is spiked with SN-38. Such behavior was earlier published with SN-38 loaded on human serum albumin (HSA) [71]. This indicates that significant amounts of free SN-38 are not present in the material, confirming the removal of the free material by dialysis.

The reported value for loading efficiency of CPT-11, a metabolic precursor of SN-38, was ~65%, which is slightly higher but still very close to the values for SN-38 we obtained. We presume that this value may be increased if desired by varying the conditions of loading. Knowing that SN-38 is 100–1000 times more potent than CPT-11, we consider this level adequate for the in vivo application. Both lignin and SN-38-lignin nanoparticles were stable for over a week in water, while SN-38 nanoparticles, probably due to their larger size, showed signs of precipitation within 24 h. The zeta-potential is within the range of −30 to +30, which is considered optimal for biological applications. The detailed HPLC method and more results are in Appendix A.

### 2.2. The Radiolabeling

The radiolabeling, performed by the chloramine-T-based iodo-beads method, resulted in a simple and effective incorporation of the radionuclide ^131^I. It is visible on the radio image of the thin-layer chromatography strip (Figure 3A), showing waste majority of the radioactive material retained at the start point, indication bound ^131^I, where routinely over 90% of radio-material was incorporated as determined by gamma-counter of the TLC strip slices. ^131^I was chosen as a radiotherapeutic based on its excellent decades-long record of successful use in radiotherapy. Possessing relatively soft radiation, with an average energy of beta-decay of 606 keV, it penetrates the tissue moderately deep, in a range of a few millimeters, thus assuring the effect on the applied place only, sparing the healthy tissue. Moreover, we already tested ^131^I in an iron oxide-based nanomaterial applied intratumorally with good results [17,18,21]. Based on the positive results we earlier achieved with the applied 3.70 MBq dose and knowing that SN-38 is a proven strong cytostatic, here we used 1.85 MBq, which is just half of the dose, to be able to observe the interaction of these two therapeutics.

### 2.3. Biodistribution

One of the very important properties of the radiolabeled nanomaterial is fate after administration by the desired route. Generally, it is desirable to have the active agent localized as much as possible in the tumor while maintaining decreased exposure of the surrounding healthy tissue and other organs to the agents. It is even more important in the case of radiotherapeutics because radiation expresses its action regardless of compartmentalization. To achieve that goal, we opted for intratumoral injection, despite intravenous injection, as a bolus or slow infusion, being the more predominant route of application of antitumor agents.

Namely, an intravenous injection inevitably exposes the whole body to the agent, frequently resulting in unwanted effects that may be limiting factors for the therapy’s success. Moreover, as a characteristic of nanomaterials, when injected i.v., it is commonly very efficiently uptaken by the reticuloendothelial system (RET), where the Kupfer cells in the liver as well as intestine uptake ~90% of the material of ~100 nm in size if injected intravenously, while the RET system in the lungs uptakes the larger (~3 µm) particles. Consequently, a very limited amount of 5–10% remains in circulation long enough to have a chance to reach the tumor. Further, solid tumors’ morphology (presence of stroma, irregular blood supply lines) and physiology (high interstitial pressure, altered pH) frequently create repulsive mechanisms that prevent the material from circulation entering the tumor. All of this results in very limited uptake of the material by tumor cells, frequently in the range of 0.2% or lower, which is insufficient for antitumor effect yet toxic to the rest of the body. This is even truer with radioisotope-containing material, where the radiation acts indiscriminately on all the places exposed and can cause serious unwanted effects.

On the contrary, we showed that iron oxide-based nanomaterial injected intratumorally frequently stays well localized in the place of injection, with minimal leakage [17]. In the last decade, there have been increasing attempts to develop antitumor therapy based on intratumoral injection of small drugs, antibodies, and biopolymers.

With all of this in mind, we investigated how the planned dose of nanomaterial behaves after the intratumoral injection. Because cancer is a disease that affects both genders pretty equally, for general pharmacokinetic study in vivo, we included both genders. But because here we investigated breast cancers, which predominantly are female-specific, the detailed study was conducted on females only. To do so, we injected 3 male and 12 female Balb/c mice with the planned dose of ^131^I-lignin or ^131^I-SN-38@lignin intratumorally into 4T1 tumors. Three males and three females were monitored by an in vivo imager equipped with a radioactivity filter for general body distribution, while the remaining three groups of the three female mice were sacrificed on days 1, 3, and 7 to accurately measure the radioactivity in tumors and organs of interest. The results are presented in Figure 3 and Appendix A.

As expected, it was found that the largest quantity of the material remained in place after the injection, with over 90% staying intratumoral 24 h after the injection. The high intratumoral level lasted as long as seven days, with minimal leakage to the organs. The only exception was the thyroid and salivary gland, which showed a relatively high accumulation of the radioactive material. Knowing the native affinity of the thyroid gland to iodine and that iodine-radiolabeled material usually undergoes enzymatic as well as non-enzymatic de-iodination in vivo, we presume that this was the case with lignin-based iodine. Another possibility is that unconjugated (free) iodine remained in the material, contributing to the level in thyroid and salivary glands. To ameliorate this potential problem, a purification step before the injection, alone or combined with an injection of a symporter in the form of Lugol’s iodine solution, could alleviate this problem. This behavior will be closely investigated to fully understand its nature and take steps to alleviate it as much as possible. Also worth noticing is that the level in the liver progressively increased to ~10% within seven days. Other than that, the lignin-based nanomaterial behaved as expected, showing prolonged retention in the place of injection.

Even though it is still a bit uncommon, intratumoral injection is gaining more and more popularity. The most obvious advantage, if compared to other routes, is a localized effect with a lack of systemic toxicity as well as simplicity. This route of administration is being extensively studied not only for small molecules and nanomedicines but also for other therapeutics, including biomolecules and immunotherapeutic agents [72,73,74,75,76,77]. We believe this application route will gain more popularity in tumor treatments in times to come.

### 2.4. Antitumor Effect

Even cancer is a disease that affects both genders pretty equally. Some tumors are sex-specific (prostate cancer for males, ovary or uterus for females), while some are biased (98% of breast cancer affects females while only 2% males). Here, we investigated the antitumor properties of ^131^I-lignin-SN-38 nanomaterial in a brachytherapy approach on a model of mouse breast cancer in female mice. The 4T1 mammary carcinoma is a transplantable mouse breast cancer cell line (of stage IV triple-negative breast cancer human equivalent) that can spontaneously metastasize to the lymph nodes, liver, lung, brain, blood, and bone while the primary tumor is growing in situ. To investigate the antitumor effect of ^131^I- and/or SN-38-loaded lignin-based nanoparticles (NPs) as a promising candidate for cancer treatment, subcutaneous xenografts of firefly luciferase-expressing 4T1 tumors were treated with the nanomaterial by intratumoral injection in a nanobrachytherapy approach. The growth of the tumors was monitored for 14 days and compared to that of non-treated or lignin-only-treated controls. The growth curves of individual tumors are presented in Appendix A. Simultaneously, the body mass of the animals was recorded as a sign of general toxicity.

It was observed that the non-treated and lignin nanoparticle-treated tumors grow at a similar rate, showing an exponential growth rate during the time observed (Figure 4A). The tumors treated with 1.85 MBq/250 µg of ^131^I-lignin showed a moderate decrease in the growth rate, resulting in twofold smaller tumors at day 14 if compared to non-treated tumors. It is in concordance with our previous results obtained with a dose of 3.7 MBq of ^131^I applied on iron oxide-based nanomaterial to treat human LS174 colon cancer xenografts on NOD-SCID mice [21]. Further, the tumors treated with 25 µg SN-38 in 250 µg of lignin as SN-38@lignin resulted in a much better antitumor effect, with the final tumor sizes being 3.5-fold smaller than non-treated ones. And finally, the therapy by 1.85 MBq of ^131^I and 25 µg SN-38 as ^131^I-SN-38@lignin resulted in the best antitumor effect, with the final tumor sizes being sixfold smaller than non-treated ones. Moreover, neither non-treated nor treated animals showed any significant weight loss, suggesting the absence of major general toxicity effects (Figure 4B). Most of the ^131^I-lignin and ^131^I-SN-38@lignin tumors still showed radioactivity presence at day 14 when excised (Figure 4C,D).

The statistical analysis of the antitumor effect by two-way ANOVA shows that, while the control and lignin-treated groups are not statistically different, proving that lignin itself does not cause any significant antitumor effect, there is highly significant statistical difference (*p* << 0.01) of the averages of calculated volumes of all the three treated tumor groups (Figure 4A) as well as the final masses of the excised tumors at day 14 (Figure 4E). Individual paired *t*-tests show that the combined effect is significantly different from both the control and lignin-treated groups, indicating its effectiveness.

To more closely investigate the effect of the therapy on the tumors, the samples of the tumors after the therapy were snap-frozen in OCT, cryosectioned, and stained by H&E for pathohistological examinations. Macroscopically, the 4T1 tumors were firm, solid, regularly shaped, and well localized. The pathohistology observation of the thin-layer slides revealed the presence of spindle-shaped tumor cells and discrete focal infiltration of leukocytes (Figure 5). Compared to non-treated ones, lignin-only injected tumors show no significant changes, while both ^131^I- and SN-38-lignin-treated tumors showed prominent fields of necrotic and apoptotic cells. The most prominent pathohistological change found in all the ^131^I-SN-38-lignin-treated tumors was massive necrosis, observed in almost 50% of the surface of all interpreted slides. This finding confirms the effects of radiation and SN-38 on a cellular level, which correlates well with the suppression of the tumor growth rate observed macroscopically.

The excellent antitumor effects obtained by the combination of ^131^I-generated radiotherapy and SN-38-mediated chemotherapy, presented here, may be reasonably explained by the mechanisms of action of radiation and camptothecins. Namely, SN-38 is one of the most active drugs from the camptothecin class, which is, in fact, an active metabolite of CPT-11, the prodrug already in human use as a first-choice therapy for colon cancers. Knowing that camptothecins act as poisons of DNA-repairing enzyme Topomerase-I as well as that radiation, here generated by ^131^I, causes the damage to DNA, we expected complementary action of these two agents, resulting in a stronger antitumor effect. The absence of general toxicity was obviously due to the lack of notable exposure of the normal organs to significant amounts of toxic materials.

## 3. Material and Methods

Alkali (kraft) lignin used to generate nanoparticles was from Sigma-Aldrich and was used without further treatment. Anticancer compound SN-38 was from MedChemExpress, New Jersey, USA. Luciferin was from GoldBio, St Louis, USA. Dimethyl sulfoxide (analytical grade) and Acetonitrile (99.99% for HPLC) were from Fluka, Buchs, Switzerland. RPMI-1640 tissue culture medium and trypsin/EDTA solution were from Sigma, while fetal bovine serum (FBS) of South American origin was from Capricorn Scientific, Ebsdorfergrund, Germany. TissueTek OCT for tissue embedding was from Sakura Finetek Japan Co, Ltd., Tokyo, Japan. The source of ^131^I in form of Na [^131^I] dissolved in NaOH solution at >37GBq/mL, was purchased from Institut National Des Radioelements, Avenue de l’Espérance, B-6220 Fleurus (Belgium). Iodination beads were from Pierce Biotechnology, Rockford, IL, USA, purchased through Thermo Scientific, Waltham MA, USA. and used as per the manufacturer’s protocol. Superfrost plus microscopic slides for frozen tissue were from New Erle Scientific LLC, Portsmouth, NH, USA. All the tissue culture plasticware was from BioRad, Hercules, USA. The common laboratory acids, bases, and salts were from Applichem-Panreach, Darmstadt, Germany, and were used as is, without purification.

The 4T1 mouse breast cancer cells were from ATCC, further engineered to express an enzyme luciferase from firefly in the laboratory of Dr. Steve Roffler, IBMS Academia Sinica, Taipei, Taiwan R.O.C. The female Balb/c mice of 8–10 weeks and ~20 g were from Vojno-Medicinska Akademija, Belgrade, Serbia. The tissue was cryosectioned on a cryotome from Keda Instruments, China.

### 3.1. Preparation and Analysis of the Nanoparticles

The nanoparticle production was based on solvent replacement methods published earlier [40,78]. Briefly, pure lignin or SN-38 solution and their mixture were prepared in DMSO in 5 mg/mL for lignin and/or 1 mg/mL for SN-38. The solutions were filtered through 0.22 µm syringe filters and transferred into prepared dialysis tubes with a cut-off of 14 kDa. The dialysis was carried out 2 × 2 h against water pH 5.2 acidified by 1 N HCl. After that, the dialysis was carried out against pure water 2 × 10 h. The dialysates were transferred to centrifuge tubes and centrifuged for 30 min 15,000× *g*. The pellet was re-dissolved in the desired solvent (water or saline) for size and SN-38 content analysis or in vivo application. Further, for FTIR analysis, the suspensions were freeze-dried 24 h to generate powder form.

The hydrodynamic radii of the nanoparticles were measured in aqueous colloidal dispersions at 25.0 ± 0.1 °C using a Nano ZS90 (Malvern, UK) instrument equipped with a 4 mW He-Ne laser source (λ = 633 nm).

The surface chemistry of the particles was investigated using attenuated total reflectance Fourier transform infrared (ATR-FTIR) spectroscopy in the mid-infrared region (4000–400 cm^−1^). A Nicolet iS50 FTIR spectrometer equipped with a Smart iTR ATR sampling accessory was used for the analysis. Powdered samples were pressed onto a diamond internal reflection element (IRE) using a swivel press to ensure good contact and efficient light interaction. Background spectra were collected from a clean, dry diamond crystal and subtracted using OMNIC™ Spectra Software 9.2.98 (Thermo Fisher Scientific, Waltham, MA, USA).

### 3.2. HPLC Analysis

The content of SN-38 in the nanomaterial was determined by an inline solid phase extraction RP-HPLC system we originally developed for the analysis of CPT-11 and its metabolites in biological material [68]. The SPE-HPLC system is assembled from two Waters 626 pumps and controllers, a µBondapack SPE and a µBondapack analytical column, a Gilson Valvemate 6-port valve, a Waters UV/Vis spectrophotometer detector and the JASCO FP-980 fluorescence detector, all controlled by a N2000 HPLC data logging system. The method successfully separates and concentrates SN-38 from lignin and other impurities on an SPE column under low acetonitrile concentration, followed by the transfer of the trapped SN-38 to the analytical column and its analysis under high-acetonitrile conditions in the high-salt and low-pH mobile phase. Briefly, the dry powder containing free or lignin-bound SN-38 was dissolved in DMSO, followed by an appropriate serial dilution (commonly four successive tenfold dilutions) in the low-acetonitrile mobile phase (5% ACN in 0.025 M K-phosphate pH-2.9) and injected into the HPLC system equipped with 100 µL loop running the mobile phase at 1 mL/min by pump 1 and the 6-port valve at position 1, directing the liquids through UV/Vis detector set up to 280 nm. After 2.5 min (the time well enough to separate and wash away the lignin and impurities), the 6-port valve was switched to position 2, which re-directs flow of pump 2, running the analytical mobile phase (26% ACN in 0.1 M K-phosphate pH = 2.9) in the opposite direction through the SPE column, dissolving the analytes and carrying them further to the analytical column for separation, as detected on the fluorescence detector set to excitation of 375 nm and emission at 540 nm. The spectra were acquired, integrated, and analyzed by an N2000 HPLC data logging system.

### 3.3. The Radiolabeling

To efficiently radiolabel the lignin, an iodo-beads method based on chloramine-T chemistry was employed according to the manufacturer’s manual. Briefly, a radioactive ^131^I in the form of NaI was diluted by PBS (10 mM phosphate, 0.9% NaCl) pH 6.8 to form a stock solution of 370 MBq/mL. A total volume of 100 µL of the stock containing the dose of 37 MBq was added in four 25 µL portions to a micro-centrifuge tube containing a single iodo-bead and exposed for 45 s when the solution was transferred to a second micro-centrifuge tube containing 1 mg/0.1 mL of lignin-based nanomaterial in PBS. After the completion of the procedure, the bead itself was transferred into the reaction and kept for an additional 10 min with occasional flicking. The reaction was terminated by the removal of the liquid over the bead. The radiolabeling efficiency was determined by TLC on the paper strips employing water pH = 5.2 as a mobile phase by imaging the radioactivity on Bruker Xtreme II bioimager or counting the radioactivity on consecutive 1 cm long strip fragments on a Perkin-Elmer Wizard II 2470 counter.

### 3.4. The Tumors Establishment

The animals were kept in cages of an Allenstown positive/negative ventilated cage system, fed by a standard laboratory rodent palleted dry food, and drank water acidified by HCl to pH = 2.8 to control the intestinal flora (*Pseudomonas* species). Access to the water and food was *at libitum*, under an artificial 12 h/12 h circadian cycle at a temperature of 23–25 °C in the accredited animal facility of the Department of Radioisotopes, Vinca Institute. Mice were maintained, and experiments were conducted in accordance with the local regulations approved by the Ministry of Agriculture, Forestry, and Water Economy, approval no. 002087449-2024-14841-002-001-323-002.

4T1 mouse breast cancer cells expressing firefly luciferase were cultured in RPMI-1640 tissue culture medium supplemented by 10% (*v*/*v*) of fetal bovine serum and appropriate antibiotic penicillin/streptomycin 100 IU/mL and 100 µg/mL, respectively, kept in an atmosphere of 5% CO_2_ at a temperature of 37 °C. The cells were split by trypsinization every 2–3 days when ~80% confluent. On the day of harvest, the cells were detached by trypsin/EDTA, washed 2 times by RPMI-1640 without the serum, and re-suspended into the medium at 5 × 10^6^ cells/mL. One hundred microliters containing 5 × 10^5^ cells/mL were injected subcutaneously by a 1 mL syringe with a no. 25 needle into a shaved right flank of Balb/c mice. The growth of the tumor was observed after day 4, and the size of the tumors and body mass of the animals were measured after day 6. When the average tumor size reached ~100 mm^3^ (day 8), the animals were ready for the therapy.

### 3.5. The Nanomaterial In Vivo Localization, Retention, and Tissue Distribution

To preliminarily determine the localization, retention, and distribution of the nanomaterial, the three female and three male tumor-bearing mice with tumors ~200 mm^3^ were injected intratumorally with 250 µg of ^131^I-SN-38@lignin containing 1.85 MBq of ^131^I and 25 µg SN-38 (Appendix A). On days 1, 3, and 7, the anesthetized animals received 3 mg/mice solution of luciferin and were photographed in Bruker Xtreme II in vivo imaging system in three modes: visible, luminescent, and radioisotopic. The radioactivity intensity in tumors in vivo was integrated by built-in software, and the time dependence of the values was displayed as uncorrected and decay-corrected data. Further, to accurately determine the tissue distribution, 9 female 4T1 tumor-bearing mice were injected as before and sacrificed at days 1, 3, and 7. Organs of interest and tumor were removed, accurately weighted, and the radioactivity measured in a Perkin-Elmer Wizard 2480 gamma-counter. The decay-corrected values in organs of interest are displayed, taking the injected dose as 100%.

### 3.6. The Antitumor Effects Determination

For the therapy, the female tumor-bearing mice were grouped into five groups containing 7–8 mice each. The first group received no treatment, serving as a control for normal tumor growth rate. The second group received 250 µg of lignin nanoparticles in 50 µL of saline. The third group received 250 µg of ^131^I-radiolabeled lignin containing 1.85 MBq ^131^I. The fourth group received 250 µg of lignin containing 25 µg SN-38. The fifth group received 250 µg of ^131^I-SN-38@lignin containing 1.85 MBq of ^131^I and 25 µg SN-38. The conditions of the animals and the growth of the tumors were observed daily while the measurement of the tumor size in three dimensions by a caliper, as well as the animal’s body mass measurement by a scale, was carried out every two days. The tumor volume was calculated by formula (*a* × *b* × *c*)/2 where *a*, *b*, and *c* were the dimensions in millimeters. The individual tumor growth is presented in Appendix A.

At the end of the experiment (day 14 or if the tumor overgrowth 2500 mm^3^ or the animal’s health score increases to an unacceptable level), the animals were sacrificed by cervical dislocation, and the tumors and the organs of interest were surgically removed. The tumor and organs were weighted, and the radioactivity was measured by Perkin-Elmer Wizard 2480 gamma-counter. The tumors were accurately weighted and photographed for the records. The correlation between the calculated final tumor volumes and masses of the surgically removed tumors is presented in Appendix A. Further, for histopathology analysis, the tumors were embedded into OCT, snap-frozen, and kept at −80 °C until needed.

### 3.7. Histopathology

For the histopathology, the OCT-embedded tumors were cut out to 5 µm sections on a cryotome and dried on RT on a superfrost plus slides. The sections were re-hydrated by water, fixed by fresh 4% paraformaldehyde, stained by a standard hematoxylin–eosin stain, de-hydrated by a toluene replacement solvent, and then mounted in Entelane. The prepared slides were observed under a Zeiss AxioVert A1 microscope, Carl Zeiss, Oberkochen, Germany, at 200× and 400× magnification and photographed by Zeiss AxioCam, Carl Zeiss, Oberkochen, Germany, with special attention to the interpretation of the degree of necrosis in the tumor. The observation and analysis of the slides were performed by a professional pathologist.

## 4. Conclusions

The research demonstrated that a combination of drugs from the camptothecin family, here presented by SN-38, with radiotherapy, here mediated by ^131^I radionuclide, applied in the form of a nanomaterial via an intratumoral injection in a nanobrachytherapy approach is indeed a promising safe and effective way to treat solid tumors. Using lignin as a safe, economical, simple, abundant, and probably biodegradable material as a potential nanocarrier for other drugs and radioisotopes may be a valuable platform for the further development of this promising approach.

## Figures and Tables

**Figure 1 pharmaceuticals-18-00177-f001:**
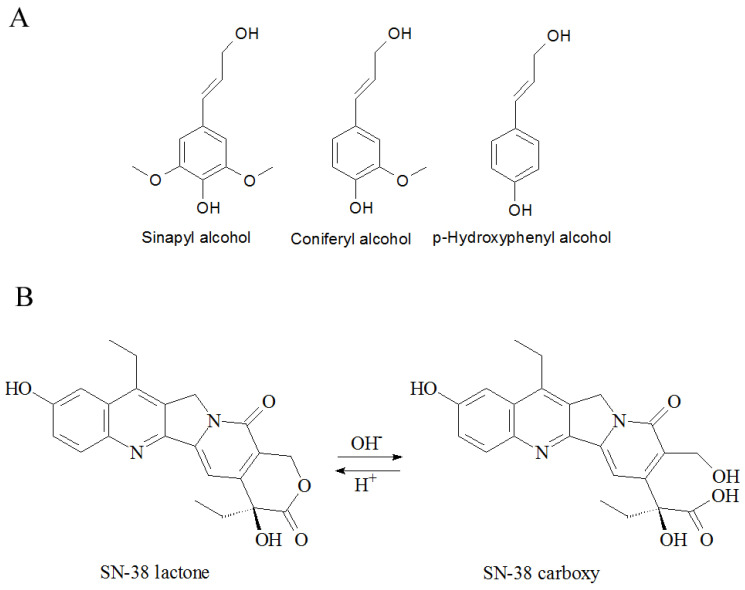
(**A**). Structure of lignin building blocks, (**B**). The structure of SN-38 and pH-dependent lactone–carboxy equilibrium.

**Figure 2 pharmaceuticals-18-00177-f002:**
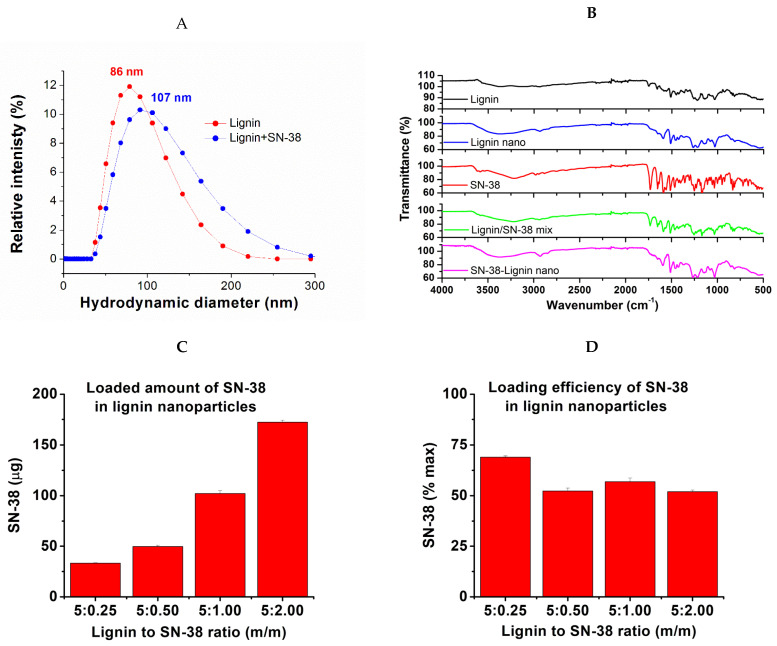
Preparation and characterization of the lignin-based nanomaterial. (**A**). Size distribution by dynamic light scattering. (**B**). FTIR spectra of the lignin, SN-38, and SN-38@lignin. (**C**). SN-38 content in SN-38@lignin nanomaterial. (**D**). Loading efficiency of SN-38 in lignin.

**Figure 3 pharmaceuticals-18-00177-f003:**
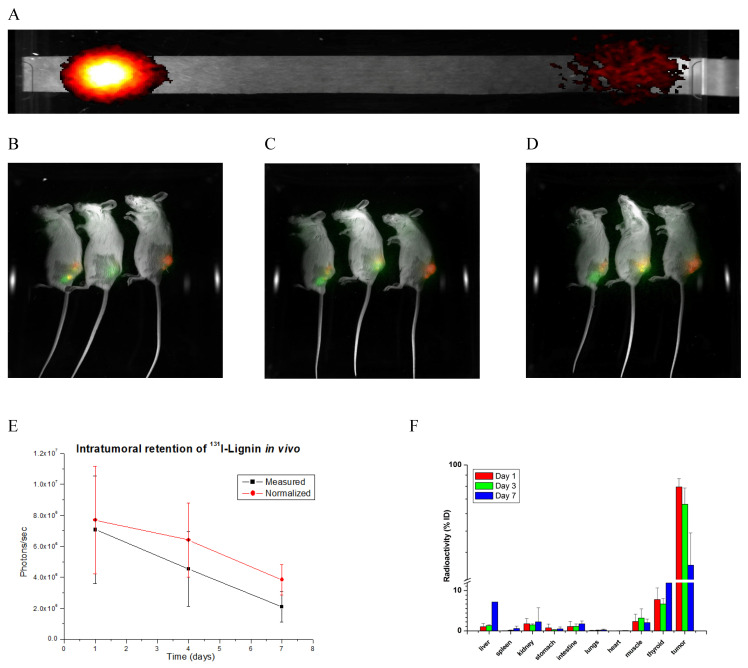
Behavior of the radiolabeled ^131^I-lignin nanomaterial in vivo. (**A**). TLC of radiolabeled material ^131^I-lignin. (**B**–**D**). In vivo imaging of established tumors (red) and intratumoral radiation (green) retention of ^131^I-lignin, and yellow representing the co-localisation at days 1, 3, and 7 after the nanobrachytherapy. (**E**). Intratumoral retention of the material determined from the in vivo imaging presented as integrated values (black line) and the values normalized for radioactive decay (red line). (**F**). Tumor and solid tissue distribution of the radiation at days 1, 3, and 7.

**Figure 4 pharmaceuticals-18-00177-f004:**
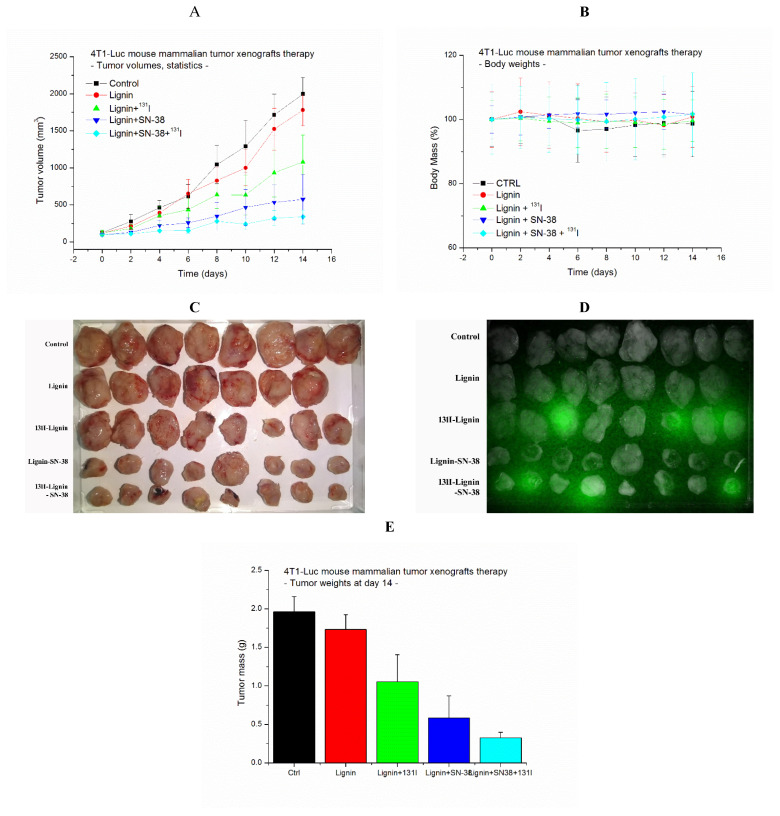
The antitumor effect of ^131^I-SN-38@lignin on 4T1 tumor xenografts on Balb/c mice. (**A**). Graphs of the tumor growth by time. (**B**). Body mass of the animals. (**C**). Photo and (**D**). radiation image of the excised tumors on day 14. (**E**). The final tumor masses after the surgical removal. The values present the averages with standard deviations, *n* = 8, except lignin (red), where *n* = 7.

**Figure 5 pharmaceuticals-18-00177-f005:**
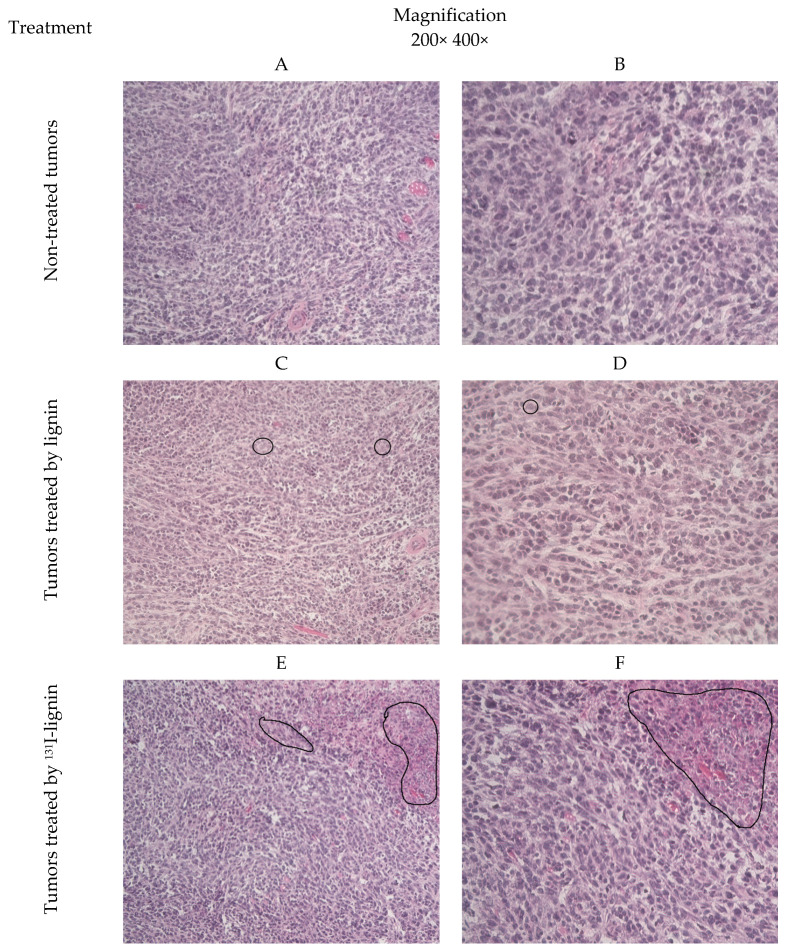
Pathohistology analysis of the 4T1 tumors excised on day 14 of the therapy. Non-treated (**A**,**B**), lignin-treated (**C**,**D**), ^131^I-lignin-treated (**E**,**F**), SN-38-lignin-treated (**G**,**H**), and ^131^I-SN-38@lignin-treated tumors (**I**,**J**) are OCT-embedded, snap-frozen, and thin-layer sectioned at 5 µm. The thin-layer slides stained by H&E, mounted in Entellan, observed at 200× (**left column**) and 400× (**right column**) magnification.

## Data Availability

Data is contained within the article or Appendix A.

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
