# Peer review of "Lignin-Based Nanocarrier for Simultaneous Delivery of 131I and SN-38 in the Combined Treatment of Solid Tumors by a Nanobrachytherapy Approach"

_pharmaceuticals, 2025, doi:10.3390/ph18020177_

Round 1
Reviewer 1 Report
Comments and Suggestions for Authors
The paper titled “Lignin-Based Nanocarrier for Simultaneous Delivery of 131I and SN-38 in the Combined Treatment of Solid Tumors by a Nanobrachytherapy approach” has discussed the combination of chemotherapy and brachytherapy in terms of breast cancer. There are a few concerns which must be addressed before publication of this article:
1. The first paragraph of Section 3 is unnecessary, as it is already covered in the Introduction part.
2. The size variation in Table S1 seems a lot from batch to batch and hence very high standard deviation of both lignin NP and with SN-38.
3. What does y-axis signify in Figure 2A?
4. Please annotate the peaks in Figure 2B and discuss the significance of each peak in the results section.
5. Please explain the huge variation in signals in Figure 3E.
6. Express Fig 3F in terms of %ID (Injected dose). It is difficult to appreciate the difference in terms of %Radioactivity.
7. Why are there different tumor numbers in Fig 4C?
8. Why is the fluorescence variable in different tumors in Figure D. If possible, provide better resolution image
9. Provide p-values for Fig 4 and E
10. The presence of nanoparticles as shown by highlights in Fig 4 is not appreciable. Please provide fluorescence images.
11. It is not clear what the authors are trying to focus on Fig 5-whether they are trying to show leukocytes or apoptotic cells or necrotic cells. For better representation, please stain with specific markers and provide quantitative values using softwares like ImageJ.
Comments on the Quality of English LanguageSatisfactory but there are grammar errors and typos in the manuscript which needs to be taken of.
Author Response
Reviewer 1:
The paper titled “Lignin-Based Nanocarrier for Simultaneous Delivery of 131I and SN-38 in the Combined Treatment of Solid Tumors by a Nanobrachytherapy approach” has discussed the combination of chemotherapy and brachytherapy in terms of breast cancer. There are a few concerns which must be addressed before publication of this article:
We do appreciate the effort to help us improving the paper. The answers on the questions and explanations are given below.
Q1. The first paragraph of Section 3 is unnecessary, as it is already covered in the Introduction part.
A1. The paragraph removed, one introductory sentence retained, as follows:
“Here, we present the synthesis, characterization and anticancer properties of lignin-based nanomaterial laden with potent cytostatic SN-38 and radioisotope 131I applied in a NBT approach against 4T1 mouse tumor xenografts.”
Q2. The size variation in Table S1 seems a lot from batch to batch and hence very high standard deviation of both lignin NP and with SN-38.
A2. Yes, this is absolutely right. Even the size variations within the same batch are fairly narrow and PDI around 0.2 which is very good, the variations in size averages among batches are notable, ie. the reproducibility was not the best. We attributed it very probably to the variations in pH of water during the dialysis step, which controls the size and is difficult to keep constant. However, all the batches produced the material which is suitable for in vivo application, so we did not investigate deeper. The matter is additionally discussed further in the text as follows:
“Even the nanoparticle size distribution we obtained is fairly narrow within the batch and well below 250 nm which is considered adequate for biological applications, there were a significant variation in the average size (but not in PDI index) among batches. We attribute it most probably to the difficulties to precise control the pH of water during the dialysis, a critical step in the procedure. However, every batch had the average size below 250nm and PDI~0.2, meaning that every batch could be used in vivo.”
Q3. What does y-axis signify in Figure 2A?
A3. The Fig.2A represents result of the size distribution measured by Dinamic Light Scattering. The Y-axis represents a percentage of particles with certain size, the X-axis represents the size while the numbers typed on the graph represent the average values of the size (not identical with the maximal peak value, which represents most abundant size!). For clarity, the new figure 2A created, with Y-axis label retained “Intensity (%)” as the original instrument’s exit plot denotes.
Q4. Please annotate the peaks in Figure 2B and discuss the significance of each peak in the results section.
A4. Trying not to diverge too much in the narration, we originally refrained from fully annotating the lignin and SN-38 FTIR spectra. However, understanding that it may be of an importance here, the data are presented in new Fig.2B graph and part of chapter 3.1. describing FTIR is completely re-done. Namely, because the SN-38 presence is fully confirmed and quantitated by SPE-HPLC analysis which cannot distinguish a mixture from incorporated SN-38, the FTIR spectra of SN-38-Lignin were recorded with intention to prove that there is no free SN-38 present. To demonstrate this, a new FTIR spectra of a lignin spiked with the same amount of SN-38 is included, where the SN-38 characteristic lines were present together with lignin spectra. Absence of new lines in SN-38-Lignin FTIR indeed confirmed that there is no free SN-38 in the material so here we discussed it in this matter. A similar behavior is observed and described earlier as well, in SN-38-HSA nanoconjugates, so this is cited as well.
Q5. Please explain the huge variation in signals in Figure 3E.
A5. The fig.3E represents the integrated values of tumor-based radioactivity-induced light emission in vivo from pictures 3B,C and D, recorded by Bruker Xtreme II in vivo imager. Being an indirect method and dealing with several factors characteristic for in vivo imaging (tissue penetration and fur caused light scattering of light as well as the tissue penetration of 131I radiation which is few millimeters only) as well as the characteristic of the instrument itself (filter-based conversion of the radiation to light and lack of collimator), the data can (and frequently do) vary considerably. Here they serve barely to roughly illustrate the localization and the elimination. However, the advantage is that the measurement is done in the same animals during the whole period with no need to sacrifice them, so very few animals (just 3) is used to get the rough idea of the dinamics. This approach is used as a preliminary test to determine does the material and technique shows enough promising potential to justify the further use. The detailed and far more accurate values (but far more animals sacrificed as well) are done with tissue distribution analysis of organs of sacrificed animals and use of gamma-counter Perkin-Elmer Wizard2 as presented in next panel F of the figure 3. It is worth noticing that, despite the variations, the averages recorded in tumors in vivo correlate very well with the data obtained from excised tumors.
Q6. Express Fig 3F in terms of %ID (Injected dose). It is difficult to appreciate the difference in terms of %Radioactivity.
A6. It is indeed already Radioactivity (%ID), but the “injected dose” was erroneously omitted. The typo corrected.
Q7. Why are there different tumor numbers in Fig 4C?
A7. The group size for in vivo antitumor therapy is calculated for every tumor type before the experiment. It is a compromise between a need to allow satisfactory confidence of the statistical analysis and need to keep the number of animals as low as possible so obeying to 3Rs law recommendation for animal use. For 4T1 and its growth characteristics, minimally 6 and maximally 8 animals is considered adequate. We originally planned 7 animals per group. However, due to tumor induction success rate being no 100%, it is common to inject ~10% mice more, to get enough tumors of required size at the moment of therapy. It is particularly difficult to plan the growth of tumors to inject it at a time point (usually 8-10 days) before the therapy, so precisely to coincide with the 131I radiation dosing, which naturally decayed during the nanomaterial preparation and characterization. It happens that our tumor induction rate was better than expected so in the moment of experiment we had few more tumor-bearing mice with right size tumors than we planned. To maximize the use of the animals, we opted to include as much of animals as available, which results in more confidentiality of the data obtained. A difference in groups size is of no concern in such cases, ie. there are no benefits of uniform (but smaller) group sizes.
Q8. Why is the fluorescence variable in different tumors in Figure D. If possible, provide better resolution image.
A8. We presume the panel in question is 4D, right? It is not fluorescence, but a visible light generated on a filter exposed to irradiation from excised tumors after the therapy. The green color resembling fluorescence is a user-defined and can be changed among 4 the instrument offerrs, but is a bit of standard in the field to use green for radioactivity. Being designed to in vivo imaging, the picture possesses limitations the same as described in Q5 (penetration through the tissue, lack of collimator, to name just two). The camera of the instrument is an actively cooled CCD of an exceptional sensitivity, but of limited resolution (5MP only). The dots appearing are not a poor resolution but are the grains of the crystals in the filter used to record, usually binned in clusters (from 3x3 to 12x12) to get the signal. Their spreading out of tumor borders is due to lack of collimation and the nature of radiation to stray aside. Sorry, it cannot be improved, because the original is presented as is. Per our knowledge there is no in vivo imager with better resolution camera available, and even if there is it was out of our reach.
Q9. Provide p-values for Fig 4 and E.
A9. We presume Fig4A and E are in question, right? Yes, the statistical analysis was included and the p-values are added. Sorry for the obvious omission. Further, a paragraph with the comments added as follows:
“The statistical analysis of the antitumor effect by two-way ANOVA shown that, while the control and lignin treated groups are not statistically different, proving that lignin itself does not cause any significant antitumor effect, there is highly significant statistical difference (p<<0.01) of the averages of calculated volumes of all the three treated tumor groups (Fig. 4A) as well as the final masses of the excised tumors at day 14 (Fig. 4E). Individual paired t-tests shown that the combined effect is significantly different from both control and lignin-treated group, indicating its effectiveness.”
Q10. The presence of nanoparticles as shown by highlights in Fig 4 is not appreciable. Please provide fluorescence images.
A10. We presume it still refers to Fig 4D as in Q5 and Q8, right? We presume the Fig 4D was misunderstood as a fluorescent image due to the green light present, but in fact it is a visible light generated by a conversion of radiation striking a filter. We direst to A8 for more details. As there is no fluorescence mentioned nor recorded anywhere in this figure, nor the nanoparticles are fluorescent enough to be visible in vivo, we admit we are a bit confused what the reviewer asked here, so cannot answer more specifically.
If the comment refers for the need for the nanoparticles localization by another method than radioactivity, we are out of luck as well. Because there is no available stain for lignin, we attempted to visualize the nanoparticle presence in excised tumors on tissue sections by trying to develop an original staining for lignin by floroglycinol, based on a method available for plant tissue. Unfortunately, the harsh conditions required for the reaction (concentrated HCl) ruins the tumor samples. We must say that this way of visualization as shown in Fig 4D is best available with the instrumentation we have.
Q11. It is not clear what the authors are trying to focus on Fig 5-whether they are trying to show leukocytes or apoptotic cells or necrotic cells. For better representation, please stain with specific markers and provide quantitative values using softwares like ImageJ.
A11. Because the macroscopic effect of the therapy on the tumors, in form of the growth suppression and the tumor sizes and masses are sufficiently documented in Fig 4, and the mechanism of the effect was not investigated, our goal with the tissue sections was barely to demonstrate that there are changes on microscopic level as well. This was demonstrated by H&E staining where zones of necrosis are clearly visible, as we did in most of our recently published papers dealing with the same topic. Because we are searching for best nanobrachytherapy material, here we are focused on antitumor effect only, as it is stated in the title. The deeper investigation, which is of utmost importance once we chose the best material and optimize the dose and route of application, will be necessary so to investigate the mechanism of the action. We are especially interested in the immune system involvement in the response, aiming for in situ vaccination effect. However, with this material, being our first therapy approach, the therapy doses and regimes still being not tested, we did not plan to go so deeply. Hence we hope the histopathology sections can stay as is. If not, they can be moved to Supplementary or omitted if the reviewer considers them bearing not important information or inappropriate in the recent form.
Comments on the Quality of English Language
Q1. Satisfactory but there are grammar errors and typos in the manuscript which needs to be taken of.
A1. The text was re-checked for grammatical errors and typos.

Reviewer 2 Report
Comments and Suggestions for Authors
In this manuscript (pharmaceuticals-3416242), the authors present the development and characterization of a biocompatible lignin-based nanomaterial that carries both the potent camptothecin-class drug SN-38 and the radiotherapeutic 131I. This dual-agent system is designed for nanobrachytherapy to deliver radionuclide and chemotherapy simultaneously to tumors. The antitumor efficacy of this nanomaterial was tested on luciferase-expressing 4T1 mouse breast cancer xenografts in Balb/c mice. Before considering publication, the authors should address the following comments:
- What specific interactions between SN-38 and lignin contribute to the consistent loading efficiency (~55%) across a broad range of SN-38 concentrations, and why does this efficiency increase to 65% at lower SN-38 levels?
- What is the zeta potential of the prepared nanoparticles: lignin nanoparticles, SN-38+lignin nanoparticles, and SN-38+lignin+131I nanoparticles?
- What functional groups or characteristic peaks in the FTIR spectra confirm the successful loading of SN-38 onto lignin nanoparticles? Additionally, please label any shifts or changes in the peaks that indicate interactions between lignin and SN-38 in the FTIR data (Figure 2B).
- How does the retention of radioactivity in the tumors treated with 131I-lignin and 131I-SN-38@lignin by day 14 correlate with the observed antitumor effects, and what implications does this have for long-term treatment efficacy?
- What specific advantages does intratumoral injection have over intravenous injection in terms of minimizing systemic exposure and maximizing tumor localization of the radiolabeled nanomaterial, particularly for radiotherapeutic agents?
Author Response
Reviewer 2:
In this manuscript (pharmaceuticals-3416242), the authors present the development and characterization of a biocompatible lignin-based nanomaterial that carries both the potent camptothecin-class drug SN-38 and the radiotherapeutic 131I. This dual-agent system is designed for nanobrachytherapy to deliver radionuclide and chemotherapy simultaneously to tumors. The antitumor efficacy of this nanomaterial was tested on luciferase-expressing 4T1 mouse breast cancer xenografts in Balb/c mice. Before considering publication, the authors should address the following comments:
We do appreciate the effort to help us improving the paper. The answers on the questions and explanations are given below.
Q1. What specific interactions between SN-38 and lignin contribute to the consistent loading efficiency (~55%) across a broad range of SN-38 concentrations, and why does this efficiency increase to 65% at lower SN-38 levels?
A1. Because the loadings were suitable to in vivo application, we did not investigate much deeper. However, we do presume it is a result of the competition of lignin-loading rate from one side and lactone-carboxy conversion characteristic for camptothecins as well as to SN-38 from another. Being applied as a lactone (water-insoluble) form, SN-38 may undergo fast (within 1-2 hours) shift to pH-dependent equilibrium with carboxy (charged, water soluble) form. Carboxy form can be (and probably is) dialyzed away. Just the remaining lactone form probably retained bound to lignin. At lower SN-38 concentrations, probably there is faster binding due to lot of unsaturated binding places at lignin, so resulting in bit higher loading efficiency (but lower total amount)! To this logic points the finding that other camptothecins (CPT-11, 10-hydroxucamptothecin) do behave almost the same. On a suggestion of another reviewer, two new references are cited to support this opinion.
Q2. What is the zeta potential of the prepared nanoparticles: lignin nanoparticles, SN-38+lignin nanoparticles, and SN-38+lignin+131I nanoparticles?
A2. We indeed recorded the zeta-potential of both lignin and SN-38 in water. The values were ~-80 for lignin and – 36 to -39mV for SN-38. Knowing that the zeta-potential of lignin heavily depends on the source of lignin (type of the wood used to prepare it, the method of preparation etc), but is commonly about -30mV and that the situation with SN-38 is even more complex: existing as a lactone (uncharged) and carboxy (negatively charged) form in dynamic equilibrium which heavily depends on pH and time exposed to, it may vary considerably. Hence, the zeta-potential is of no particular use here, drifting considerably over short time. Further, once injected in animal, it interacts with the biomaterial and local pH as well. For those reasons we opted not to investigate those data closer, because they represent no relevant values for in vivo behavior.
Q3. What functional groups or characteristic peaks in the FTIR spectra confirm the successful loading of SN-38 onto lignin nanoparticles? Additionally, please label any shifts or changes in the peaks that indicate interactions between lignin and SN-38 in the FTIR data (Figure 2B).
A3. The SN-38 presence in lignin nanomaterial is strongly demonstrated and quantitated with SPE-HPLC. However, as HPLC could be positive even if SN-38 is simply present as free form but not loaded in lignin, FTIR here is applied mainly to prove there are no SN-38 peaks which indicate free SN-38 presence. To better compare the loading, we added a new FTIR spectra of lignin spiked with the same amount (10% m/m) as in nanomaterial. Further, the main peaks of both lignin and SN-38 are annotated. Finally, even the peaks of SN-38 are clearly visible in the micture FTIR, no common peaks are found in the nanomaterial despite absolutely positive confirmation of the loading by HPLC. Finding no common peaks between free SN-38 and lignin-bound SN-38, we concluded there is no free SN-38 but only the lignin-bound one. Similar behavior was published already with nanomaterial containing SN-38 and HSA. The FTIR was not recorded for 131I-SN-38-lignin due to the contamination problem.
Q4. How does the retention of radioactivity in the tumors treated with 131I-lignin and 131I-SN-38@lignin by day 14 correlate with the observed antitumor effects, and what implications does this have for long-term treatment efficacy?
A4. This is a logical question which requires a bit complex answer. Namely, the intratumoral therapy applied as NBT in most cases results in fairly narrow deposit zone. Our previous results with 131I-labeld nanomaterial visible on tissue section, shows that the killing zone is mostly in near vicinity of the deposit zone. However, within the time, the radioactivity decayed and the tumor grows, heavily interfering with the correlation of the effect with the dose and localization applied. To fairly answer the question, one should analyze the data course of the effect, ie. sacrificing few animals every few days during longer time and analyzing the distribution and the effect correlation. This being not done, the correlation at day 14 may be obscured with the uneven growth of the tumor and physiological processes, so we refrain from trying this. Consequently, we still cannot predict the best course and outcome of eventual repeated therapy.
However, the tumor growth curve indicates that the dose may be repeated after 7-10 days for a better effect. Absence of the toxicity is an additional encouraging fact for longer therapy.
Q5. What specific advantages does intratumoral injection have over intravenous injection in terms of minimizing systemic exposure and maximizing tumor localization of the radiolabeled nanomaterial, particularly for radiotherapeutic agents?
A5. Thank you for raising this important question. In fact, we were long time supporters of intravenous therapies, as most common way of anticancer medicine administration. However, during the time, facing abundance of our own and others experimental data, the intratumoral injection attracted more and more attention, till we completely switched to this route. The detailed comparison was published in our previous papers as well as in many other. Here we try not to repeat ourselves and others so just basics are incorporated in the paper. However, considering that readers may not be familiar with the popularity intratumoral injection gained last decade, we included a brief comment in Introduction as follows:
“Moreover, intratumoral injecton is steadily gaining popularity last decade, with plethora of materials, from small drugs to biopolimers to nanomaterial being delivered this route.”
And in the Results, chapter 3.3, while the biodistribution was discussed, there is already very detailed elaboration of pros and cons of intratumoral injection.

Reviewer 3 Report
Comments and Suggestions for Authors
Aleksandar Vukadinović et al reported the development and characterization of lignin-based nanocarrier for simultaneous delivery of radionuclide 131I and cytostatic SN-38. And the in vivo biodistribution and antitumor efficacy were evaluated in 4T1 mouse breast cancer xenografts on Balb/c mice. However, I think this manuscript needs major revision and is not suitable for publishing in Pharmaceuticals in its current form. Here are some comments of this manuscript for authors.
1. There are too many paragraghs in the “Introduction” section, the authors should re-organize this section in a brief manner. The section 3.1 also should be reorganized.
2. In Table S1, The standard deviations of the hydrodynamic diameters of lignin and lignin-SN-38 nanoparticles as determined by DLS are too large (nearly 50% of the size), please check.
3. There is no related description about figure 2A-C in section 3.1, the authors should describe the results and give some discussions. For example, how did the authors confirm the successful loading of SN-38 into nanocarriers? And the drug loading content should be given.
4. Similarly, the authors should describe the TLC results in figure 3A.
5. Why did the authors use the intratumoral injection in the antitumor study? It is not easy to operate and difficult in practical use. In addition, the dose of SN-38 should be given in the manuscript.
6. Some previous reports about hydroxycamptothecin delivery (DOI: 10.1002/chem.201200765, DOI: 10.1021/am503359g) should be cited in the manuscript to compare the drug loading content of different nanocarriers.
7. The writing style of the volume units is not standard, such as ml should be mL, μl should be μL. There are some typos in this manuscript, such as stomack in figure 3F should be stomach. There is an overlap in figure 4E, Lignin+SN-38 and Lignin+SN-38+131I.
Author Response
Reviewer 3:
Aleksandar Vukadinović et al reported the development and characterization of lignin-based nanocarrier for simultaneous delivery of radionuclide 131I and cytostatic SN-38. And the in vivo biodistribution and antitumor efficacy were evaluated in 4T1 mouse breast cancer xenografts on Balb/c mice. However, I think this manuscript needs major revision and is not suitable for publishing in Pharmaceuticals in its current form. Here are some comments of this manuscript for authors.
We do appreciate the effort to help us improving the paper. The answers on the questions and explanations are given below.
Q1. There are too many paragraghs in the “Introduction” section, the authors should re-organize this section in a brief manner. The section 3.1 also should be reorganized.
A1. Agreed, there were too many paragraphs in the Introduction. This was a result of the complexity of the material used for multimodal therapy. Namely, it was necessary to describe the problem, the recent approaches and familiarize the readers with every aspect of the multimodal therapy separately as well as the reason for the combination. However, we do agree that we were a bit too broad. So the whole Introduction was shortened, the paragraphs combined where appropriate and overall the text made more fluent to read.
Q2. In Table S1, The standard deviations of the hydrodynamic diameters of lignin and lignin-SN-38 nanoparticles as determined by DLS are too large (nearly 50% of the size), please check.
A2. Yes, this is absolutely right. Even the size variations within the same batch are fairly narrow and PDI around 0.2 which is very good, the variations in size averages among batches are notable, ie. the reproducibility was not the best. We attributed it very probably to the variations in pH of water during the dialysis step, which controls the size and is difficult to keep constant. However, all the batches produced the material which is suitable for in vivo application, so we did not investigate further. The matter is additionally discussed further as follows:
“Even the nanoparticle size distribution we obtained is fairly narrow within the batch and well below 250 nm which is considered adequate for biological applications, there were a significant variation in the average size (but not in PDI index) among batches. We attribute it most probably to the difficulties to precise control the pH of water during the dialysis, a critical step in the procedure. However, every batch had the average size below 250nm and PDI~0.2, meaning that every batch could be used in vivo.”
Q3. There is no related description about figure 2A-C in section 3.1, the authors should describe the results and give some discussions. For example, how did the authors confirm the successful loading of SN-38 into nanocarriers? And the drug loading content should be given.
A3. Because the SN-38 presence in the SN-38-lignin nanomaterial was undoubtfully confirmed and quantitated by SPE-HPLC analysis, the FTIR spectra of SN-38-Lignin were recorded to prove that there is no free SN-38 present. Namely, when a mixture of the same proportion of SN-38 and lignin was recorded (we added the new FTIR depicting that), the SN-38 characteristic lines were present. Absence of new lines in SN-38-Lignin FTIR indeed confirmed that there is no free SN-38 in the material so here we discussed it in this matter. This behavior is observed and described earlier as well, in SN-38-HSA conjugates, so this is cited. The content of the SN-38 in lignin was presented in panels C and D, in form of total (C) and relative (D) amount. However, respecting the fact that two reviewers asked a similar question, we included much more detailed FTIR analysis in the description.
Q4. Similarly, the authors should describe the TLC results in figure 3A.
A4. The Fig3A was not commented much because it is a visual demonstration only, while the accurate data, obtained by Perkin-Elmer Wizard2 gamma-counter, were presented in the text. However, to improve the readability, the TLC photo was commented as follows:
“It is visible on the radio image of the thin-layer chromatography strip (Fig. 3A), showing waste majority of the radioactive material retained at start point, indication bound 131I, where routinely over 90% of radio-material was incorporated as determined by gamma-counter of the TLC strip slices.”
Q5. Why did the authors use the intratumoral injection in the antitumor study? It is not easy to operate and difficult in practical use. In addition, the dose of SN-38 should be given in the manuscript.
A5. Briefly, we do agree it is not an ideal route, but is much more comfortable and easier to perform than the classical brachytherapy which it aims to replace, greatly enhancing the applicability. However, considering that many may not be familiar with the popularity intratumoral injection gained last decade, we included a brief comment in Introduction as follows:
“Moreover, intratumoral injecton is steadily gaining popularity last decade, with plethora of materials, from small drugs to biopolimers to nanomaterial being delivered this route.”
And in the Results, chapter 3.3, while the biodistribution was discussed, there is a very detailed elaboration of pros and cons of intratumoral injection.
In fact, we were long time supporters of intravenous therapies, as most common way of anticancer medicine administration. However, during the time, facing abundance of our own and others data, the intratumoral injection attracted more and more attention, till we completely switched to this route. The detailed comparison was published in our previous papers as well as in many other.
The dose of the therapy was given from the Abstract onward as 250mg lignin carrying 1.85MBq (50uCi) of 131I and 25ug SN-38 per 100mm3 tumor.
Q6. Some previous reports about hydroxycamptothecin delivery (DOI: 10.1002/chem.201200765, DOI: 10.1021/am503359g) should be cited in the manuscript to compare the drug loading content of different nanocarriers.
A6. Thankyou for pointing to those valuable papers. We were aware only on CPT-11 loading data, which ere cited already. However, 10-OH-CPT is structurally close to SN-38 so the data are relevant. We included them in the papers and commented the loading as follows:
“Interestingly, almost the same loading efficiency was reported earlier with 10-hydroxy-camptothecin, resulting in ~55% efficiency and 9% (m/m) ratio [70,71], further supporting the opinion that the loading efficiency is related to the structure of camptothecins and their lactone-carboxy equilibrium”.
Q7. The writing style of the volume units is not standard, such as ml should be mL, μl should be μL. There are some typos in this manuscript, such as stomack in figure 3F should be stomach. There is an overlap in figure 4E, Lignin+SN-38 and Lignin+SN-38+131I.
A7. We do apologize for the errors. The new Fig. 3F and 4E are generated with the corrections made and the typos and other omissions are corrected through the text.

Round 2
Reviewer 3 Report
Comments and Suggestions for Authors
The authors addressed most of my concerns.